# Non-zero transverse single spin asymmetry of very forward $\pi^0$ in polarized $p + p$ collisions at $\sqrt{s} = 510$ GeV

**Minho H. Kim[1,2⋆] for the RHICf Collaboration**

**1** RIKEN BNL Research Center, Brookhaven National Labratory,
Upton, New York 11973-5000, USA
**2** Korea University, Seoul 02841, Korea

⋆ jipangie@korea.ac.kr

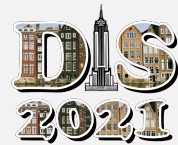 *Proceedings for the XXVIII International Workshop on Deep-Inelastic Scattering and Related Subjects, Stony Brook University, New York, USA, 12-16 April 2021*

## Abstract

The RHICf experiment measured transverse single spin asymmetry of very forward ($\eta > 6$) $\pi^0$ from polarized $p + p$ collisions at $\sqrt{s} = 510$ GeV. In order to measure it precisely, we installed a new electromagnetic calorimeter at zero-degree area of the STAR experiment at the Relativistic Heavy Ion Collider (RHIC) and measured the $\pi^0$s over the kinematic range of $x_F > 0.25$ and $0 < p_T < 1$ GeV/$c$ in June 2017. A clear non-zero asymmetry was observed even in low $p_T < 1$ GeV/$c$ showing a similar $x_F$ dependence with that of the forward ($2 < \eta < 4$) $\pi^0$. A possible diffractive contribution may need to be taken into account to explain the very forward $\pi^0$ asymmetry. RHICf-STAR combined analysis and follow-up experiment will give a clue to understand it qualitatively.

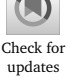 Check for updates

## 1 Introduction

Transverse single spin asymmetry ($A_N$) is a powerful observable for understanding the spin-involved production mechanism in the polarized $p + p$ collision. It is defined by a left-right cross section asymmetry,

$$A_N = \frac{\sigma_L - \sigma_R}{\sigma_L + \sigma_R},\tag{1}$$

where the $\sigma_{L(R)}$ is the cross section of a particle produced in the left (right) side with respect to the beam polarization.

Large non-zero asymmetries in the $\pi^0$ production have been discussed only in the quarks and gluons' degrees of freedom. It has been measured in a wide range of collision energies

[1–5] and these results have been explained by transverse momentum dependent (TMD) [6–8] and higher twist functions [9–11] in an initial or final state with the transverse motions of quarks and gluons.

Recently, further analysis results [12–14] showed a strange behavior in the forward $\pi^0$ asymmetry. Bigger asymmetry was observed in more isolated $\pi^0$ event which is connected to the diffractive process. In this analysis, the $\pi^0$ was considered as isolated one when its energy fraction to the detected electromagnetic particles was close to 1. In this case, since the $\pi^0$ carries larger energy fraction, the asymmetry nature could be biased. However, it also gives us a possibility of the diffractive contribution to the $\pi^0$ asymmetry because it is also expected that there is few particles around it when it is produced by the diffractive process, thereby the $\pi^0$ asymmetry induced by the diffractive process may be observed in this analysis. This means the non-zero asymmetry could be correlated with not only the partonic interaction but also the diffractive one.

In order to study the diffractive contribution to the $\pi^0$ asymmetry, the RHICf experiment measured the $A_N$ in the very forward $\pi^0$ production. The diffractive process is expected to dominate when the produced particle is measured in the very forward region.

## 2 RHICf Experiment

We installed an electromagnetic calorimeter (RHICf detector) which had been originally developed for the LHCf experiment [15] in front of a STAR hadronic calorimeter, zero-degree

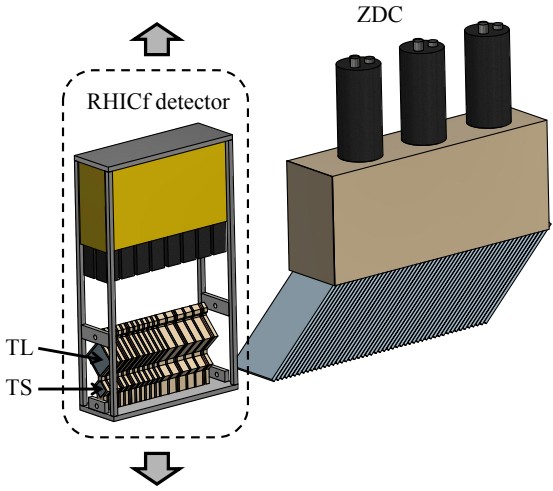

Figure 1: Schematic drawing of the RHICf detector installed in front of the ZDC. We moved the detector vertically to cover $p_T$ from 0.0 to 1.0 GeV/$c$.

calorimeter (ZDC) [16], which was located 18 m away from the beam interaction point and took the data in June 2017. Figure 1 shows the schematic drawing of the experimental setup.

The RHICf detector consists of two sampling calorimeters, the large (TL, 40 mm dimenstion) and small (TS, 20 mm dimension) towers. Both towers are composed of 17 layers of tungsten absorber with a 44 radiation lengths in total, 16 layers of GSO plate for energy measurement, and 4 layers of GSO bars. Particle energy is reconstructed by the energy deposits of the GSO plates. Position is reconstructed by the GSO bar layers, which are covered by X-Y pairs of GSO bars with 1 mm dimension.

We measured the very forward $\pi^0$ from the polarized $p + p$ collisions at $\sqrt{s} = 510$ GeV. In order to measure the wide $p_T$ coverage, we used the radial polarization which was the 90°-

rotated one than usual vertical polarization. We also moved the detector vertically so that the zero-degree direction of the beam faced the center of the TL, the center of the TS, and 24 mm below the center of the TS. We requested larger $\beta^*$ value of 8 m and lower luminosity $\sim 10^{31}$ cm$^{-2}$s$^{-1}$ than usual to make the systematic uncertainty by the angular beam divergence small. The $\beta^*$ is an indicator for how much the beam is squeezed. With these detector positions and special settings, $\pi^0$s over the longitudinal momentum fraction range of $x_F > 0.25$ and the transverse momentum range of $0 < p_T < 1$ GeV/$c$ were measured. For the measured $\pi^0$s, the RHICf detector has an energy resolution of 2.5%~3.5% and $p_T$ resolution of 3.0%~4.5%.

$\pi^0$s could be identified by measuring two decayed photons in each tower (Type-I) or both ones in same tower (Type-II). A Type-I $\pi^0$ trigger was used for the measurement of the Type-I events. It was operated when the energy deposits of three successive layers of both towers are larger than 45 MeV. Since the electromagnetic shower stops the development in the middle of the detector, only the upstream seven layers were used for this trigger. The Type-II events were measured by a high electromagnetic (high-EM) trigger. It was operated when the energy deposit of the fourth layer of either tower is larger than 500 MeV.

## 3 Data analysis

Due to the detector geometry, we used following luminosity formula to calculate the $\pi^0$ asymmetry

$$A_N = \frac{1}{PD_\phi}\Big(\frac{N_L - RN_R}{N_L + RN_R}\Big), \tag{2}$$

where $P$ is the beam polarization, $N_{L(R)}$ is the number of detected $\pi^0$s in the left (right) side of the beam polarization. $R$ is the luminosity ratio of the spin directions resulting into the events to right and left sides. It was estimated using STAR beam beam counter (BBC) [17] and vertex

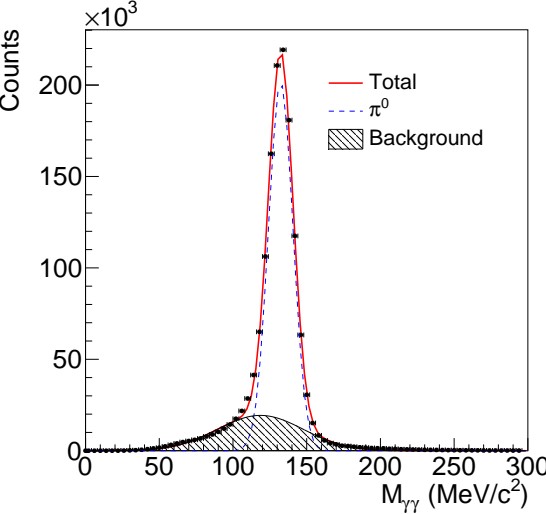

Figure 2: Reconstructed two photon invariant mass distribution of Type-I event in $x_F > 0.25$ and $0.0 < p_T < 1.0$ GeV/$c$.

position detector (VPD) [18]. $D_\phi$ is a correction factor to compensate the diluted asymmetry by the azimuthal angle distribution of the detected $\pi^0$s. Only the Type-I $\pi^0$ triggered events

were used for the asymmetry calculation of the Type-I events and high-EM triggered events for the Type-II events to be free from the effect of the different detection efficiencies.

$\pi^0$s to be analyzed were identified using invariant mass distribution of two detected photons. Figure 2 shows the reconstructed invariant mass distribution. In this distribution, a clear $\pi^0$ peak (blue dashed line) was observed in the widely smeared background events (black filled histogram). The background mostly comes from two accidental photons from different $\pi^0$s. Superposition of the Gaussian (for describing the $\pi^0$ events) and the 6th order of polynomial (for describing the background events) function was used to fit the invariant mass distribution. $3\sigma$ width of the Gaussian peak was considered as the final $\pi^0$ candidate. The effect of the background events included in the final $\pi^0$ sample was subtracted using the following equation

$$A_N^S = \Big(1 + \frac{N_B}{N_S}\Big)A_N^{S+B} - \Big(\frac{N_B}{N_S}\Big)A_N^B, \tag{3}$$

where $A_N^{S+B}$, $A_N^S$, and $A_N^B$ are the estimated asymmetries in the signal+background, signal only and background only regions. The $A_N^B$ was estimated using the events whose invariant mass was further than $5\sigma$ of the Gaussian peak and the signal to background ratio, $\frac{N_B}{N_S}$, was calculated from the fit result.

The $\pi^0$ asymmetry was calculated as functions of $x_F$ and $p_T$ to study their correlation. Systematic uncertainties by the polarization and beam center estimations, and the background subtraction were included. The effect of the smearing due to the finite $x_F$ and $p_T$ resolutions were studied by GEANT4 using the single $\pi^0$ beam which was artificially weighted to reproduce the asymmetry, but they were negligible. For each bin divided by selected $x_F$ and $p_T$ values, more than 90% of migrated events were from $\delta x_F < 0.025$ and $\delta p_T < 0.035$ GeV/$c$ of the bin boundaries. The differences between the calculated and true $\langle x_F \rangle$, $\langle p_T \rangle$, $A_N$ values of each bin were less than 0,008, 0.009 GeV/$c$, and 0.0015, respectively.

To find any missing systematic uncertainties, a "bunch shuffling" analysis was proceeded. It is done by randomizing the spin patterns and reconstructing the $A_N$ again. Since the spin patterns are randomized, ideally, the reconstructed $A_N$ should be fluctuated around 0 with its statistical uncertainty if there is no systematic source missed. The bunch shuffled asymmetries were consistent with zero in the comparable fluctuation with the statistical uncertainties. We conclude that there are no noticeable false asymmetries in this analysis.

## 4 Results

Figure 3 shows the $A_N$ of very forward $\pi^0$ as functions of $x_F$ and $p_T$. Increasing $A_N$ as a function of $p_T$ up to $\sim$0.2 is clearly shown in Figure 3 (a) and the corresponding $p_T$ range is where the diffractive process dominates. In Figure 3 (b), the backward asymmetries which are the ones in the opposite side of the polarized beam are consistent with zero. The forward asymmetry is also consistent with zero at very low $p_T < 0.07$ GeV/$c$, but it starts to increase as a function of $x_F$ as $p_T$ increases.

Figure 4 shows the comparison of the RHICf data with the forward $\pi^0$ asymmetries previously measured by FNAL, PHENIX, and STAR. It shows that the increasing asymmetries of the very forward $\pi^0$ get comparable with the forward $\pi^0$ ones at higher $p_T$ though it is still smaller than 1 GeV/$c$. The RHICf result also shows the same $x_F$ scaling with the forward $\pi^0$ ones. The diffractive process may contribute to the asymmetries at higher $p_T$ than it has been expected. Since the RHICf data may be also a tail of the partonic one, more detailed analysis than inclusive one is necessary. Correlations with STAR's central detectors and Roman pots will make it possible by identifying from which process the $\pi^0$ with non-zero asymmetry

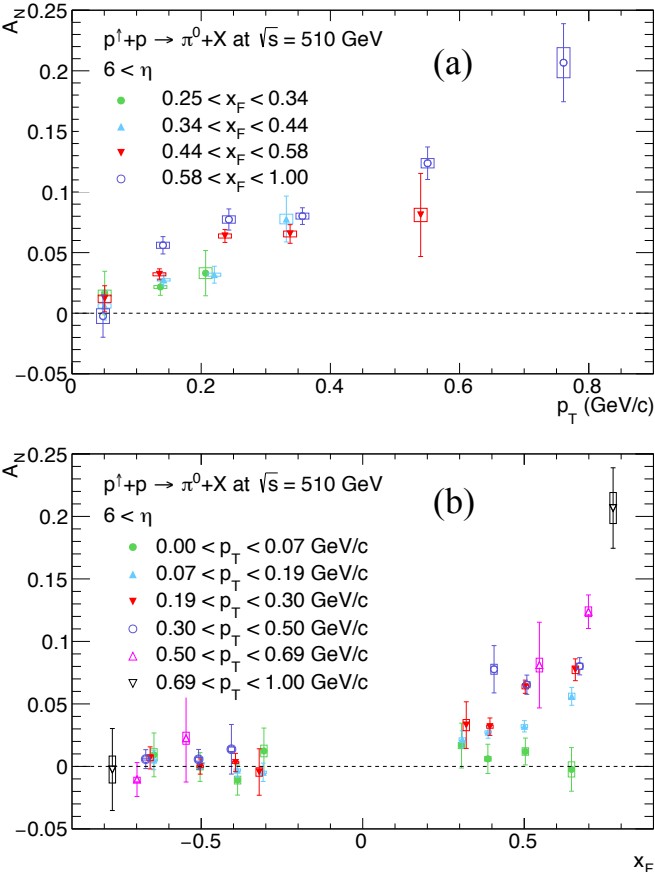

Figure 3: $A_N$ of the very forward $\pi^0$s as functions of (a) $p_T$ for several $x_F$ ranges and (b) $x_F$ for several $p_T$ ranges. Only forward $A_N$ was presented in (a). Error bars represent the statistical uncertainties, and the boxes represent the systematic uncertainties.

comes. It is also desirable to investigate the same observables in the unexplored kinematic region between 0.8∼2.0 GeV/$c$ where the fractions of both partonic and diffractive processes are comparable.

## 5 Conclusion

The role of the diffractive process to the $\pi^0$ asymmetry can be studied by measuring the $A_N$ of the very forward $\pi^0$. At low $p_T < 1$ GeV/$c$ where the diffractive process is expected to dominate, large non-zero asymmetries increasing as functions of both $x_F$ and $p_T$ have been observed for the first time from the polarized $p+p$ collisions at $\sqrt{s} = 510$ GeV. The asymmetries show an approximate $x_F$ scaling with the forward $\pi^0$ ones, which may indicate a possible diffractive contribution to the $\pi^0$ asymmetries. The RHICf result will be further investigated by RHICf-STAR combined analysis and a follow-up experiment, RHICf-II.

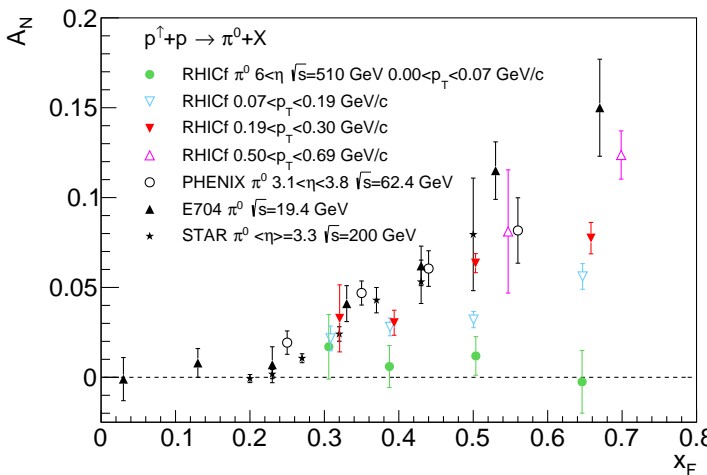

Figure 4: Comparison of the RHICf data with the previously measured $A_N$ of the forward $\pi^0$s as a function of $x_F$.

## 6 Future Prospect

In order to more deeply understand the RHICf result, RHICf-STAR combined analysis has been started. Using STAR detectors covering each $\eta$ region, the type of the production mechanism is expected to be identified.

A follow-up experiment (RHICf-II) is also being prepared. To measure more various particles and for higher statistics, a larger detector of 8 cm × 18 cm dimension is planned to be constructed. Technology of ALICE FoCal-E [19] will be transferred to the construction of the RHICf-II detector. Currently, we're optimizing the definite design of the detector and planning to take the data in 2024 at STAR.

**Funding information** This program is partly supported by the U.S.-Japan Science and Technology Cooperation Program in High Energy Physics, JSPS KAKENHI (No. JP26247037 and No. JP18H01227), the joint research program of the Institute for Cosmic Ray Research (ICRR), University of Tokyo, and the National Research Foundation of Korea (No. 2016R1A2B2008505 and No. 2018R1A5A1025563), and "UNICT 2020-22 Linea 2" program, University of Catania.

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
