# Peer review of "Non-zero transverse single spin asymmetry of very forward $\pi^0$ in polarized $p + p$ collisions at $\sqrt{s} = 510$ GeV"

_SciPost Physics Proceedings, doi:SciPost Phys. Proc. 8, 060 (2022)_

## Round 1 · Referee Report · Anonymous (Referee 1) · 2022-2-28

Report

This paper, "Non-zero transverse single spin asymmetry of very forward π^0 in polarized p + p collisions at sqrt-s = 510 GeV" describes a measurement of the transverse single spin asymmetry in very forward π^0's. The experimental apparatus is well described for a paper of this type, and the description of the data analysis is appreciated and illuminating. The results presented are mostly clearly laid out and should be published in this journal.

However there are several comments that should be addressed before this manuscript can be published in this journal.

There are several quantities that are introduced but not defined when they are first used, namely \beta^*, x_F, and p_T. I also find Fig. 4 to be a bit hard to read, specifically the label. Making this figure larger could significantly improve the readability of the figure. Finally it would be appreciated if the authors could take a closer look at the grammar used in the manuscript. In several spots throughout the document there are errors that make parsing individual sentences difficult, most notably in the first two sections.

---

## Editorial Decision

published